# CiberAMP: An *R* Package to Identify Differential mRNA Expression Linked to Somatic Copy Number Variations in Cancer Datasets

**DOI:** 10.3390/biology11101411

**Published:** 2022-09-28

**Authors:** Rubén Caloto, L. Francisco Lorenzo-Martín, Víctor Quesada, Arkaitz Carracedo, Xosé R. Bustelo

**Affiliations:** 1Molecular Mechanisms of Cancer Program, Centro de Investigación del Cáncer, CSIC-University of Salamanca, 37007 Salamanca, Spain; 2Instituto de Biología Molecular y Celular del Cáncer de Salamanca, CSIC-University of Salamanca, 37007 Salamanca, Spain; 3Centro de Investigación Biomédica en Red de Cáncer (CIBERONC), CSIC-University of Salamanca, 37007 Salamanca, Spain; 4Departamento de Bioquímica y Biología Molecular, Universidad de Oviedo, 33006 Oviedo, Spain; 5Center for Cooperative Research in Biosciences (CIC-bioGUNE), Basque Research and Technology Alliance (BRTA), Bizkaia Technology Park, 48160 Derio, Spain; 6Ikerbasque, Basque Foundation for Science, 48013 Bilbao, Spain; 7Traslational Prostate Cancer Research Lab, CIC-bioGUNE, Biocruces Bizkaia Health Research Institute, 48903 Barakaldo, Spain

**Keywords:** pan-cancer, RNA sequencing, gene expression, transcriptome, software, glioblastoma

## Abstract

**Simple Summary:**

The ability to establish accurate correlations between the number of copies of genes and the expression levels of their encoded transcripts remains a challenge despite the extensive progress made in the understanding of the genome of cancer cells. Here, we describe a new algorithm that does so by integrating both genomics and transcriptomics data from the Cancer Genome Atlas. In addition to explaining the step-by-step basis of this new method, we provide examples of how this new algorithm can help identify functionally meaningful gene copy alterations that are recurrently detected in cancer patients.

**Abstract:**

Somatic copy number variations (SCNVs) are genetic alterations frequently found in cancer cells. These genetic alterations can lead to concomitant perturbations in the expression of the genes included in them and, as a result, promote a selective advantage to cancer cells. However, this is not always the case. Due to this, it is important to develop in silico tools to facilitate the accurate identification and functional cataloging of gene expression changes associated with SCNVs from pan-cancer data. Here, we present a new *R*-coded tool, designated as CiberAMP, which utilizes genomic and transcriptomic data contained in the Cancer Genome Atlas (TCGA) to identify such events. It also includes information on the genomic context in which such SCNVs take place. By doing so, CiberAMP provides clues about the potential functional relevance of each of the SCNV-associated gene expression changes found in the interrogated tumor samples. The main features and advantages of this new algorithm are illustrated using glioblastoma data from the TCGA database.

## 1. Introduction

SCNVs are commonly found in most cancer types [1,2,3]. Since they are associated with either the gain or loss of gene dosage, it is widely assumed that such genomic alterations must drive significant changes in the expression of either proto-oncogenes or tumor suppressor genes [2,4]. However, it should be noted that these SCNV-linked differentially expressed genes (SCNV-DEGs) may not be functionally relevant, as their amplification or deletion could be the result of their physical proximity to amplified or deleted regions harboring cancer driver genes [3,5]. In addition, changes in gene dosage may not always correlate with the expected changes in gene expression [2]. While those issues can be easily approachable using wet-lab techniques when dealing with very specific SCNV-DEGs and tumor samples, they become difficult to tackle from a genome-wide and pan-cancer perspective. To solve this problem, in silico tools are needed to integrate the extensive amount of genomic and transcriptomic data currently available from ongoing cancer genome sequencing efforts.

The first bioinformatics approximations to deal with this problem were based on correlation methods to link SCNVs and gene expression changes [6,7,8,9,10,11,12,13,14,15,16]. However, the performance of such methods has left significant room for improvement mostly in terms of specificity [17,18]. Furthermore, the detection of significant SCNV connected with differential expression changes by most of these methods is highly dependent on the high levels of SCNV recurrence in a tumor dataset analyzed [19]. This feature makes it difficult to discover functionally relevant alterations that are found at low frequency in cancer patients. Another problem is that these methods only consider changes in expression levels of the interrogated genes between SCNV-positive and diploid tumor samples but not in relation to those found in healthy tissue samples. Due to this, these algorithms cannot give a comprehensive view of the variation in expression levels of a given gene in normal tissues, in diploid tumors, and when included within genomic regions that underwent SCNV events. Finally, these algorithms cannot provide direct information on the genomic context in which SCNV-DEGs take place. As a result, it is difficult to elucidate the actual functional relevance of the hits obtained. To solve these problems, a second-generation bioinformatics tool (Oncodrive-CIS) was developed [20]. However, this method still has some room for improvement. Thus, Oncodrive-CIS utilizes a system-integrated tool rather than a more modular analytic pipeline to identify SCNV-DEGs. Consequently, it is difficult for the users to autonomously modify the searching parameters to identify the SCNV-DEGs present in a given genomic dataset. It does not integrate either the information on gene co-amplification and co-deletion within the same copy number altered genomic region and, as a result, it cannot give information on whether such alterations might have proactive or bystander effects for cancer cells. Lastly, Oncodrive-CIS does not allow the user to find out if a specific gene is upregulated or downregulated in tumor samples versus normal controls. The integration of this parameter in SCNV-DEG analyses is interesting, given that the somatic copy number alterations may lead to changes in the transcriptional status of a gene that can be totally different from those found between healthy tissue and diploid tumor samples.

Here, we describe a new *R* package tool, which has been called CiberAMP, which uses differential expression analyses to establish accurate correlations between specific SCNVs and changes in the expression of the genes affected by them. Unlike other previously described methods, CiberAMP can yield information on: (i) The SCNV-DEGs present in a given cancer genome. (ii) The type of expression changes elicited by them (e.g., if they are SCNV-dependent, whether they amplify the changes already seen between healthy samples and diploid tumors, or whether they change the expression pattern relative to those found in normal tissues and diploid tumors). (iii) The genomic context in which such SCNV-DEGs occur. (iv) The potential functional relevance of each identified SCNV-DEG. To validate this new algorithm, we demonstrate here that it can be used to identify functionally relevant SCNV-DEGs using genomic and transcriptomic data from glioblastoma samples deposited in the TCGA.

## 2. Materials and Methods

### 2.1. TCGA RNA-Seq Data Download, Filtering, and Normalization

To perform these analyses, CiberAMP first downloads RNA-seq raw count data from the Genomic Data Commons server for each queried TCGA cohort using the *TCGAbiolinks R* package (version 2.25.2) [21]. Then, the expression matrix is filtered in two steps. First, an array–array intensity correlation analysis identifies outliers between samples. To that end, it calculates a square symmetric matrix of Spearman correlation coefficients among all samples and removes those associated with a value lower than 0.6 (this is an arbitrary threshold set in the *TCGAbiolinks R* package by default and can be modified by the user). Secondly, poorly expressed genes are pulled out from the analysis by calculating the average expression for each gene across all the samples and, according to the data obtained, removing those within the lowest quartile group. Finally, the filtered matrix is subjected to two normalization steps using the functions provided by the *EDAseq R* package (version 2.30.0) and is also integrated into the analytical pipeline of *TCGAbiolinks.* The first one involves a “within-the-lane” normalization to eliminate the count dependence on either the length or GC content of the interrogated genes. The second one is a “between-the-lane” normalization process that reduces the count dependence on sources of variation between samples (e.g., differences in sequencing depth). These steps are computed as described in the original paper on the normalization function [22]. Detailed information about the *R* packages utilized in CiberAMP can be found in Appendix A. Instructions for the installation of the CiberAMP package can be found on the GitHub page (https://github.com/vqf/ciberAMP/ (accessed on 25 September 2022)).

### 2.2. Differential Expression Analysis

Upon normalization of mRNA expression data, CiberAMP performs its first major step to identify all differentially expressed genes between tumor and normal samples (when such information is available in the datasets) using the *edgeR* algorithm [23,24]. The *edgeR* pipeline is first used to calculate a common dispersion for every gene within the matrix using the Conditional Maximum Likelihood method [23]. Subsequently, *edgeR* performs a pairwise gene differential expression contrast and statistically evaluates differences in mRNA levels between the two sample conditions. To this end, the algorithm applies Fisher’s exact test to calculate a P for each comparison, which is then further adjusted using the false discovery rate (FDR) method. Genes with an associated FDR value less than 0.05 and a difference in mRNA levels between the two conditions of at least 50% (log_2_(FC) > 0.58) are considered significantly deregulated by default. These parameters can be modified by the user to make more relaxed or stringent the analysis of differentially expressed genes.

### 2.3. Classification of Tumor Samples

The second major step carried out by CiberAMP is the classification of every interrogated gene as amplified, deleted, or diploid. To that end, the algorithm downloads from the Broad Institute’s *FireBrowser* server the outcomes obtained from the latest run of the GISTIC2.0 algorithm [25] on the raw copy number data of each queried cohort using the *TCGAbiolinks R* package. Specifically, it downloads the designated “thresholded by gene” file that contains a matrix of numbers that indicates the type of copy number variation per sample and per gene using the following notation: −2, which means gene homozygous deletion; −1, which refers to a gene hemizygous deletion; 0, which indicates a diploid content; +1, which indicates the gain of an extra copy of a given allele; and +2, which refers to a high-level gene copy gain. Importantly, CiberAMP allows the integration using either shallow (−1 and +1) or deep (−2 or +2) copy number variations to be performed as well as the possibility of analyzing them in a single group.

### 2.4. Copy Number and mRNA Expression Data Integration

Upon execution of the above steps, CiberAMP performs the differential expression analysis between copy number altered and diploid tumor samples for every gene queried. Subsequently, it interrogates the resulting list of differentially expressed genes to assess if the queried gene is among them. As a result, CiberAMP can identify the loci whose copy number variation is associated in a statistically significant manner with changes in gene expression. This leads to the generation of a final table containing all the information associated with each of the analyzed cohorts. Such information includes: (i) Differential expression of interrogated genes between tumor vs. normal as well as between SCNV altered vs. diploid tumor samples. (ii) Prevalence level of SCNVs. (iii) TCGA IDs associated with each analyzed SCNV-associated differentially expressed gene.

### 2.5. Analysis of Co-Amplified and Co-Deleted Genes

The file thresholded by genes from the Broad Institute’s *FireBrowser* data server is further encoded in binary notation where a value of 1 denotes the presence of an SCNV in a sample and a value of 0 indicates a diploid genotype. The algorithm then calculates for each pair of genes a *p* value using Fisher’s exact test and differentiates mutually exclusive and co-occurring events (co-amplifications or co-deletions) within a cohort. Finally, it returns all significant associations between SCNV-DEGs (*p* value < 0.05) with any SCNV-associated differentially expressed gene that has been cataloged as a cancer driver by the COSMIC Cancer Gene Census (https://cancer.sanger.ac.uk/census#cl_search (accessed on 21 July 2022)).

### 2.6. Identification of SCNV-DEG Enriched Genomic Regions

To reveal enriched genomic clusters, the algorithm calculates the number of SCNV-DEGs within all ten-megabase-long fragments in which the tumor genome can be split. Then, the average number across all fragments containing more than two SCNV-DEGs is calculated. Finally, fragments with a number of lodged SCNV-DEGs higher than average are considered enriched SCNV-DEG genomic sites.

### 2.7. Classification Algorithm

First, the complete set of SCNV-DEGs is stratified into two subsets depending on whether they are significantly co-amplified or co-deleted with any known cancer driver included in the COSMIC Cancer Gene Census. Second, these two subsets are further subdivided, taking into consideration whether the genomic coordinates of the gene are within or outside any of the previously calculated enriched genomic clusters. Finally, in each of the four resulting subgroups, genes are ranked from the highest to the lowest position according to SCNV recurrence and the FDR-adjusted differential expression value associated with SCNV.

### 2.8. Data Visualization

CiberAMP integrates a function to interactively explore the output of the analysis. This representation is built on a *ShinyAPP* that uses the *shiny* and *plotly R* packages to create the visualization. The rest of the plots found in this article were created using the ggplot2 *R* package [26]. Sankey diagrams were generated using the *networkD3 R* package (https://cran.r-project.org/web/packages/networkD3/index.html (accessed on 21 July 2022)) and the circular representations of the human genome were represented using the *circlize R* package [27].

### 2.9. Analysis of the Genetic Dependence of Cancer Cell Lines on Gene Amplification

The genetic dependence scores (Chronos scores) and copy number variation data of 1045 tumor cell lines were downloaded from the Dependency Map database (DepMap 21Q3 Public + Score, Chronos score) [28]. Cell lines associated with gene copy number scores higher or equal to 2.5 were classified as amplified, while those associated with scores between 1.5 and 2.5 were classified as diploid. Then, we compared the distribution of dependence scores between both cancer cell subgroups using the Wilcoxon test and calculated the difference between the medians of both distributions. The knockout of a given gene was considered to have a significant negative impact on the proliferation of cancer cell lines when associated with a median difference > 0.10 and a *p* value < 0.05.

## 3. Results and Discussion

### 3.1. Features of the CiberAMP Package

The *R* package described here, which will be referred to hereafter as CiberAMP, combines a pipeline of genomics and differential gene expression analyses to identify SCNVs associated with statistically significant changes in mRNA expression levels. CiberAMP can access and process the information from the TCGA, currently, the largest available database as it contains genomic and transcriptomic paired data from more than 10,000 patients and 33 different cancer types [29]. The main differences between CiberAMP and the previously described Oncodrive-CIS method are summarized in Table 1.

CiberAMP requires three data inputs from the users to work: (i) The list of genes to be queried (in official symbols). This list can range from a single gene to a whole set of genes of the human genome. (ii) The set of tumors to be interrogated (using TCGA cohort IDs as input) (Appendix A). (iii) Parameters to carry out the search process, including *p* value, minimum fold-change expression changes, and minimum SCNV frequency thresholds. These parameters can be those offered by default by the system or those chosen at will by the users). To process these data, CiberAMP downloads the following information from online repositories: (i) Copy number data from the latest run of the GISTIC2.0 algorithm [25] for each of the queried tumors, which are downloaded from the Firehose data server using the *TCGAbiolinks R* package (Figure 1A). (ii) RNA-seq data in raw counts format that is retrieved from the Genomic Data Commons data portal using, as above, the *TCGAbiolinks R* package (Figure 1A). The algorithm will utilize all this information subsequently to: firstly, classify each queried gene in the tumor dataset as “amplified” (low- and high-level copy number), “deleted” (homozygous or heterozygous, also referred to as deep or shallow, respectively), or “diploid” (normal 2n DNA content) (Figure 1A, step 1); secondly, conduct a whole differential gene expression analysis between SCNV-positive and diploid tumor samples (Figure 1A, step 2); and, thirdly, to perform differential expression analysis between tumor and normal samples (in the case that such information is available at the TCGA) (Figure 1A, step 3). These differential gene expression analyses utilize *edgeR*, a robust algorithm designed to identify and quantify changes in gene expression between small, asymmetrical groups of samples [23,24].

This pipeline generates three intertwined outputs: (i) The list of differentially expressed genes between tumor and normal samples. (ii) The list of differentially expressed genes between SCNV-positive and diploid tumor samples. (iii) The catalog of loci that are shared by the foregoing lists. To facilitate the visualization and exploration of the data by users, the algorithm includes a function to create an interactive *x*/*y* plot in which the coordinates of each interrogated gene are based on the difference in expression levels between tumor and normal samples (*x* axis) as well as between SCNV-positive and diploid tumor samples (*y* axis) (Figure 1A). The user can also obtain information about each identified SCNV-DEG by clicking the appropriate dot on the plot (Appendix A). This information includes the gene symbol, the type of SCNV (amplification or deletion), the recurrence of the variation, and the tumor type it was selected from (Appendix A). Finally, the user can obtain information on whether the identified SCNV-DEG is significantly co-amplified or co-deleted with any oncogene from the COSMIC Cancer Gene Census (Appendix A) upon clicking on the appropriate dot found in the *x*/*y* representation (Appendix A).

Since SCNVs usually encompass multiple loci, CiberAMP performs two additional steps to analyze the genomic context in which each SCNV-DEG takes place (Figure 1B). Firstly, it calculates all significant co-amplifications and co-deletions of known proto-oncogenes and tumor suppressor genes using the information contained in the COSMIC Cancer Gene Census [30]. Secondly, the algorithm identifies the genomic regions that harbor a higher number of SCNV-DEGs than on average (Figure 1A,B). These genomic regions will be referred to hereafter as “enriched SCNV-DEG genomic clusters”. As a result of all these analyses, CiberAMP can eventually provide the following information to the user: (i) List of SCNV-DEGs in a given set or group of tumors. (ii) The fraction of SCNV-DEGs that are concurrently amplified and/or deleted with known cancer drivers present in the neighborhood or in faraway genomic regions. (iii) The list of SCNV-DEG located within enriched genomic clusters of the steps carried out by CiberAMP to classify SCNV-DEGs according to genomic context.

### 3.2. Identification of SCNV-DEGs in Glioblastoma Using CiberAMP

To validate CiberAMP, we first analyzed its performance using a cohort of 156 TCGA-archived glioblastoma samples that contain paired genomic and transcriptomic data. This dataset also includes eight samples from healthy brain tissue to be used as controls. When all human genome genes were interrogated using the CiberAMP default parameters (log_2_(FC) > 0.58, FDR adjusted *p* value < 0.05), we were able to identify 5166 differentially expressed genes between normal brain samples and glioblastoma (Appendix A) as well as 5812 genes associated with SCNV (considered as such the gain of more than three copies or the loss of both alleles) in at least one tumor sample. From these analyses, we also identified a total number of 316 SCNV-DEGs (Figure 2 and Appendix A). These SCNV-DEGs could be classified into three main subsets according to their expression pattern: (Subset a) Genes whose deregulation is exclusively associated with SCNV events (e.g., *PTEN* and *MYCN* in the case of SCNVs involving loss and gain of gene copies, respectively) (Figure 2, see complete gene list in Appendix A). This subclass includes 43% of the SCNV-DEGs found in glioblastoma. (Subset b) Genes in which the SCNV leads to the exacerbation of the differential expression already seen between diploid glioblastoma and healthy tissue samples (e.g., *EGFR*, *CDK6*, *MYC*, and *PDGDFRA*) (Figure 2, see list of identified genes in Appendix A). This subclass represents 38% of the SCNV-DEGs identified by CiberAMP in glioblastoma. (Subset c) SCNVs that reshape the expression pattern of the gene that is found between diploid tumors and normal samples (Figure 2, see the complete gene list in Appendix A). For example, the *AKT3*, *FGFR3*, and *KIT* transcripts usually exhibit lower levels in diploid glioblastoma samples when compared to healthy controls. However, they become upregulated in the case SCNV-positive tumor cases (Figure 2). A mirror-image scenario is seen in the case of the *CDKN2A*, *CD274*, and *CDKN2C* tumor suppressor genes, which show elevated levels of expression in tumor diploid versus normal samples and reduced levels in SCNV-positive versus diploid tumors (Figure 2). This third subclass represents 19% of all SCNV-DEGs identified by CiberAMP in this tumor type.

### 3.3. CiberAMP Benchmarking

To validate CiberAMP, we compared its performance with a previously described method that was designed using a similar rationale (Oncodrive-CIS). To that end, we analyzed the same cohort of glioblastoma samples using the default parameters of each of those methods. In terms of total SCNV-DEG numbers, we found a notable difference in the total score reported by CiberAMP (316, see Figure 3A) and Oncodrive-CIS (>2000). Importantly, 314 of the 316 CiberAMP hits were also detected in the Oncodrive-CIS results obtained. The disparity in the total number of hits between the two programs is likely due to the fact that CiberAMP has more stringent criteria than Oncodrive-CIS to generate the list of SCNV-DEGs (in terms of overall gene expression changes). Consistent with this, we have found that most of the additional SCNV-DEGs reported by Oncodrive-CIS are associated with lower gene expression levels than those identified by CiberAMP (Figure 3B). Of course, this does not exclude the possibility that they are functionally meaningful. However, in terms of subsequent wet-lab characterization studies, we believe that selecting SCNV-DEGs associated with a higher impact on gene expression will be prima facie more important. These analyses indicate that CiberAMP can provide a more accurate perspective of how a given SCNV-DEG impacts mRNA levels in the cancer cohort analyzed. Importantly, these analyses can be adjusted to the specific needs of the user to carry out more stringent or relaxed searches (Appendix A).

Another critical point in this type of algorithm is the ability to identify cancer-relevant genes in the generated list of SCNV-DEGs. We found that 9.8% of the SCNV-DEGs identified by CiberAMP in glioblastoma target proto-oncogenes (22 genes) or tumor suppressor genes (9 genes) according to the information contained in the COSMIC Cancer Gene Census database (Figure 3C,D). These genes correspond to all the gene expression subsets a (4), b (15), and c (12) that have been described above (Figure 2). The most prevalent of these SCNV-DEGs are homozygous deletions targeting the tumor suppressor genes *CDKN2A* (chromosome 9, 55% of the patients) and *PTEN* (chromosome 10, 10.1%) as well as amplifications targeting the proto-oncogenes *EGFR* (chromosome 7, 47.9%), *CDK4* (chromosome 12, 18.2%), *PDGFRA* (chromosome 4, 14.2%), *DDIT3* (chromosome 12, 10.8%), and *KIT* (chromosome 4, 10.8%) (Figure 3D,E). However, the majority of SCNV-DEGs associated with copy number-altered cancer driver genes (77%) identified by CiberAMP are found at frequencies below 10% of glioblastoma cases (Figure 3D,E). These analyses demonstrate that CiberAMP can efficiently identify functionally relevant genes independently of the frequency with which they are found in patients. Importantly, CiberAMP provides a list of hits more enriched in proto-oncogenes and tumor suppressor genes across most frequency intervals than Oncodrive-CIS (Table 2). Similar results are obtained when the performance of these two algorithms is tested using a cohort of 515 head-and-neck squamous cell carcinoma tumors downloaded from the TGCA database; although, in this case, the percentage of total cancer drivers identified is similar when using both algorithms (Table 2). However, as indicated above, it is worth noting that CiberAMP always yields SCNV-DEGs that are associated with more robust fold change levels (Figure 3B). Collectively, these data indicate that CiberAMP facilitates the selective and effective identification of SCNV-DEGs when compared to the previous benchmarking method. In addition to this, it is worth noting that CiberAMP offers analytical capabilities much more flexible for the users than previous methods (see Table 1).

### 3.4. CiberAMP Provides Information for Subsequent Hypothesis-Driven Studies on SCNV-DEGs

The functional relevance of other SCNV-DEGs that are not well-established cancer driver genes is difficult to assess unless information is available from previous publications. In this regard, an important advantage of CiberAMP is that it can provide complementary information on the genomic context in which the identified SCNV-DEGs take place (Figure 4A). This feature enables the building of hypotheses on the potential functional relevance of the genes included in those SCNV-DEGs to carry out subsequent wet-lab validation experiments. For example, we can classify the SCNV-DEGs identified in glioblastoma into three hypothetical subclasses according to the information provided by CiberAMP: (i) Predicted drivers, a subclass that contains all the SCNV-DEGs linked to known proto-oncogenes and tumor suppressor genes (see above, Figure 3E). (ii) Putative drivers, a subclass that in turn includes two types of SCNV-DEGs. On the one hand, there are SCNV-DEGs (32, 10% of total) that are not associated with concurrent copy number variations and are known as cancer driver genes (Figure 4A). This subgroup encompasses six closely located loci (*CCKBR*, *CYBR5A*, *ZNF214*, *HBB*, *EPS8L2*, and *H19*) that are usually found co-deleted on chromosome 11 (Figure 4B and Appendix A) and 26 genes that are dispersed across different chromosomes (Figure 4C and Appendix A). *PDGFA* and *FAM20C* (both located at chromosome 7) are the most frequently amplified genes of this category (4% in each case), followed by amplifications of *TKTL1* (chromosome X, 2.7%) and deletions in *RYR3* (chromosome 15, 2.7%) (Appendix A). In the case of the co-deleted cluster of chromosome 11 (Figure 4B), it is likely that at least one out of the six genes present in that region would play a tumor suppressor role. The candidates are *CYB5R2* and *CCKBR*, since they have been previously associated with suppressor-like activities [31,32,33,34,35]. On the other hand, there are SCNV-DEGs (42, 13.2%) that are concurrently amplified or deleted with cancer driver-encoding SCNV-DEGs located in different chromosomes (Figure 4A,D) (Appendix A). It is likely that the inter-chromosomal linkage of those SCNV-DEGs might indicate functional cooperativity or antagonism of proteins encoded by them in cancer cell fitness. A good example of SCVN-DEGs belonging to this subclass is the amplification of the *EGFR* gene (chromosome 7), which is usually associated with the loss of the *CDKN2A* gene (chromosome 9) (Figure 4D), and has been previously related to poor prognosis in the case of glioblastoma patients [36,37,38]. (iii) SCNV-DEGs with uncertain functional relevance, which represents the most frequent subclass (211 SCNV-DEGs, 66.8% out of the total; Figure 4A and Appendix A). These SCNV-DEGs are in genomic clusters that contain known proto-oncogenes and tumor suppressor genes such as *AKT3, CDKN2C, MDM4, NTRK1*, and *PRDM2* (chromosome 1); *KIT* and *PDGFRA* (chromosome 4); *EGFR* (chromosome 7); *CDKN2A* (chromosome 9); and *DDIT3* and *CDK4* (chromosome 12). Due to this, the genes of this subclass can represent mere bystanders or, alternatively, loci that cooperate in *cis* with the proto-oncogenes or tumor suppressor genes that are located within the same chromosomal region.

Notably, the number of potentially relevant genes generated by CiberAMP can be significantly reduced by the users by simply modifying the default parameters of the system. For example, by setting the minimal expression fold-change requirement from the default (0.5) to a higher one (2.5), the total number of hits is reduced from the initial 316 to 178 using the same glioblastoma TCGA dataset. In any case, the number of hits provided by CiberAMP is perfectly suited for functional validation using low to medium throughput approaches using shRNA interference or CRISPR–Cas9-mediated gene editing.

### 3.5. Validation of CiberAMP-Identified SCNV-DEGs

Using the a priori functional characterization criteria described above, we ended up with a list of 74 SCNV-DEGs that could be potentially relevant for glioblastoma. We followed two independent strategies to obtain further insights into their potential relevance. Firstly, we analyzed the existing literature to identify previously established correlations between candidate genes and the fitness of cancer cell lines (general or glioblastoma-specific). Using this approach, we found that 38 of those SCNV-DEGs (52% of total) had indeed been linked to pro-tumorigenic (25 genes), tumor suppression (4 genes), or pro/anti-tumorigenic (9 genes) functions in a number of tumor types (Appendix A). In particular, the role of 12 of these genes (32%) was demonstrated using glioblastoma cell lines (Appendix A). This group includes the top three SCNV-DEGs identified by CiberAMP (*PDGFA*, *FAM20C*, and *TKTL1*). As a control, 90% of SCNV-DEGs linked to cancer drivers have previously been associated with pro- or anti-neoplastic activities in glioblastoma tumors (Appendix A). This percentage can be higher since, to our knowledge, no functional studies have been conducted on *FGFR3*, *IKZF1*, and *PRDM2* in this tumor type as yet.

As a second approximation, we utilized the CRISPR–Cas9-based gene dependency database (DepMap 21Q3 Public + Score, Chronos score) to determine the impact of the deletion of CiberAMP-identified hits on the proliferation rates of 1045 independent cancer cell lines (including 43 glioblastoma cell lines). Given that these analyses are not well suited to pick up increased levels of proliferation upon ablation of suppressor genes, we focused these analyses on the 56 SCNV-associated upregulated genes in glioblastoma. This approach revealed that the ablation of 50% of those genes does negatively affect the proliferation of a wide spectrum of cancer cell lines (Figure 5A). One of these genes (*POLR1C*, which encodes the subunit C of RNA polymerases I and III), appears to be essential since its knockout severely affects the growth of all interrogated cell lines (Figure 5A). Other genes also induce a statistically significant impact on the proliferation of a wide number of cell lines (*XPO5*, *RSPH9*, *QRLS1*, *MRLP14*, *MAD2L1BP*, and *SLC35B2*) (Figure 5A, left column). Eleven of those genes (19.6%) are also important for the proliferation of at least one of the glioblastoma cell lines present in that database (Figure 5A, right column, gene symbols in red). As a control, we found using the same analysis that the knockout of 70% (19) of the 27 upregulated proto-oncogenes found within SCNV-DEGs in glioblastoma (Figure 3E) causes reductions in the proliferation of some of the cell lines included in the screening (Figure 5B). Among them, *MYC* was the one showing the most pan-cancer cell line essentiality (Figure 5B). The knockout of 12 (44%) of these proto-oncogenes also affects the proliferation of at least one of the glioblastoma cancer cell lines present in the interrogated cell collection (Figure 5B, right panel, gene symbols in red).

Finally, we used a more stringent analysis of the CRISPR–Cas9-based gene dependency database to evaluate whether some of the genes included in the SCNV-DEGs influenced the proliferation of cancer cell lines according to their amplification status. To this end, we subdivided the entire collection of 1045 cell lines included in the DepMap into the “amplified” and “diploid” subsets for each SCNV-DEG under investigation and, subsequently, calculated the difference in median dependency scores between these subgroups using the Wilcoxon test (see Methods). Using this approach, we found that the knockout of *E2F3* is associated with a significant reduction in the proliferation of the cell lines that contain SCNV-DEGs harboring that gene (Figure 5C). Three additional genes impact the proliferation at lower levels (*ID4*, *POLH*, and *TKTL1*) (Figure 5C). In the case of SCNV-DEGs harboring cancer drivers, we found eight of them whose amplification status is associated with proliferation rates in cancer cell lines (Figure 5D). Collectively, these results indicate that the information on the genomic context of SCNV-DEGs provided by CiberAMP can be used to pinpoint a significant number of genes that can influence a very specific parameter of cancer cells (effect on proliferation rates). It is likely that the spectrum of functionally relevant SCNV-DEGs could be even larger if we incorporate other experimental readouts for their characterization. Likewise, it is likely that some of them would require cooperating inputs from other SCNV-DEGs or pathobiological programs present in cancer cell lines.

## 4. Conclusions

Here, we have reported a new in silico tool, CiberAMP, which will facilitate studies aimed at the identification of SCNV-DEGs in pan-cancer data with the widely used *R* platform. This method is flexible in terms of the selection of screening parameters by the users as well as quite robust in performance according to our benchmarking tests. In addition to allowing the identification of SCNV-DEGs according to user-defined parameters, CiberAMP provides extra layers of information that may help to select candidates for further validation. These additional strata of information include: (i) The specific effect that such SCNVs elicit in the expression pattern of the targeted genes and whether such an effect is similar or different from those observed between diploid tumors and normal tissue samples. (ii) The genomic context in which such genomic alterations take place (e.g., association or not with enriched SCNV-DEG genomic clusters, or linkage to similar cancer driver SCNV-DEG located in the vicinity or faraway in the genome). With this information, users can obtain a comprehensive view of SCNV-DEGs at the whole-genome or single-gene level for any desired TCGA tumor and, perhaps more importantly, can establish hypotheses to subsequently select specific SCNV-DEGs for further functional validation using either wet-laboratory experiments or online experimental database resources.

## Figures and Tables

**Figure 1 biology-11-01411-f001:**
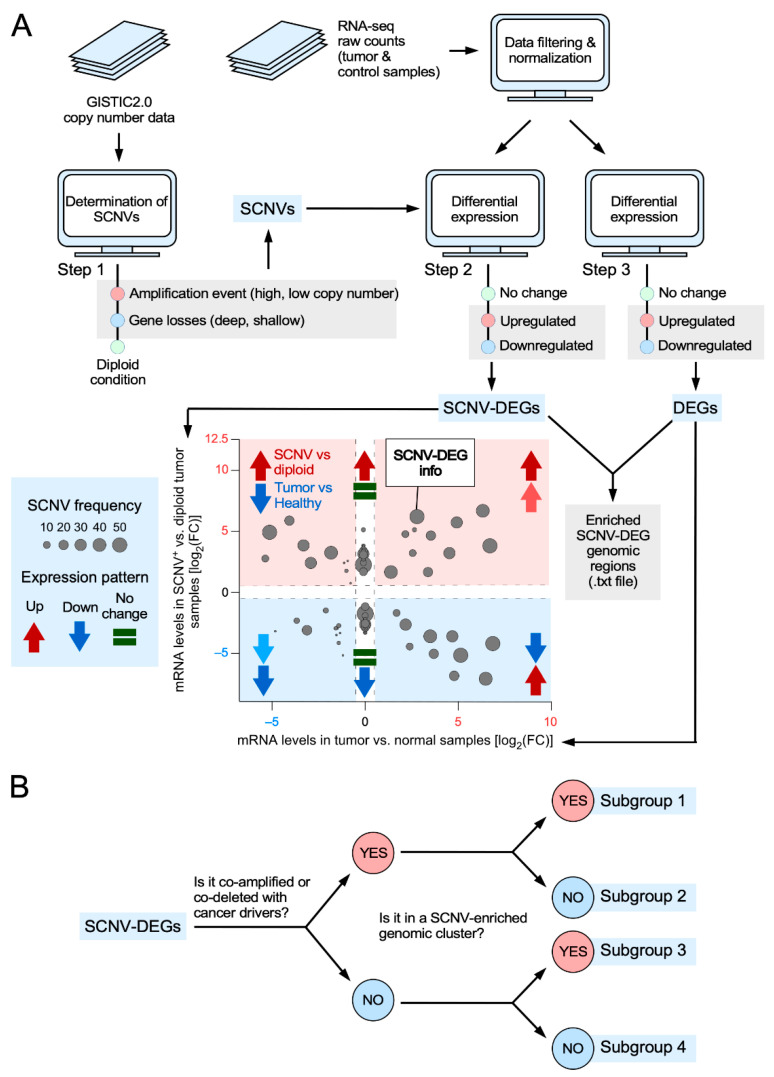
Scheme of the in silico pipeline used by the CiberAMP *R* package. (**A**) CiberAMP routine used to identify and visualize SCNV-DEGs. Icons used are explained in the blue box on the left. (**B**) Summary of the steps carried out by CiberAMP to classify SCNV-DEGs according to genomic context.

**Figure 2 biology-11-01411-f002:**
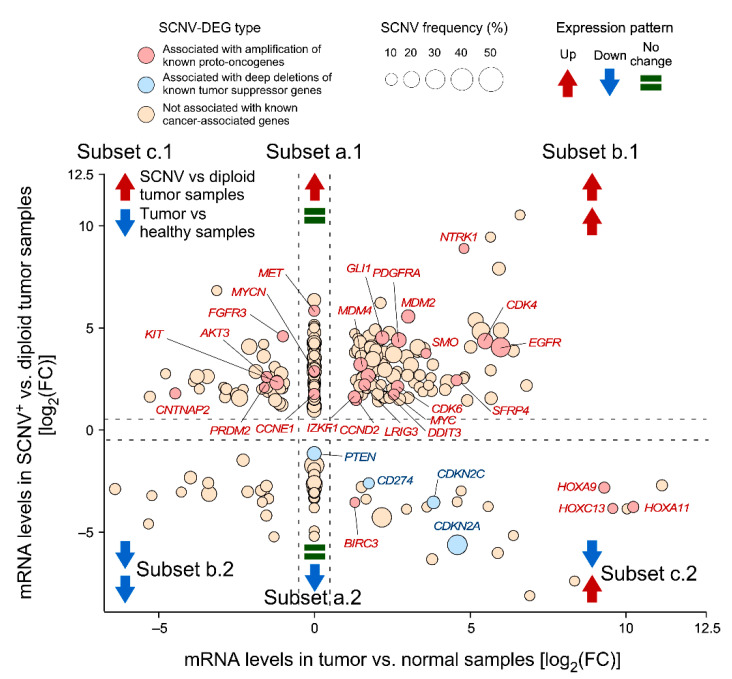
Identification of SCNV-DEGs in glioblastoma using CiberAMP. Example of the CiberAMP-generated graphic representation of candidate SCNV-DEGs using glioblastoma data from the TCGA. In addition to SCNV-DEG visualization according to amplified or deep deletion status (coordinates of SCNV-DEG within the *x*/*y* axes), information is given about the relative frequency of each individual SCNV-DEG in the tumor genomic data contained in the TCGA (size of circles). More information about each SCNV-DEG can be obtained by clicking on each point (see Appendix A). This representation also allows users to classify SCNV-DEGs according to their expression patterns in SCNV^+^, diploid, tumor, and control samples, as indicated in the main text. To facilitate the interpretation of the data by the readers, the graph has been modified to include extra information on known proto-oncogenes (red color) and tumor suppressor genes (blue color). Further information about the symbols used is shown at the top of the figure. The subsets depicted in the figure are: subset a.1, genes whose expression is only upregulated when included within an SCNV; subset a.2; genes whose expression is only downregulated when included within an SCNV; subset b.1, genes whose expression is upregulated in SCNV versus diploid samples as well as tumor versus healthy samples; subset b.2, genes whose expression is downregulated in SCNV versus diploid samples as well as tumor versus healthy tissue samples; subset c.1 and c.2, genes that show the opposite pattern of expression in SCNV versus diploid tumor samples and in tumor versus healthy tissue samples.

**Figure 3 biology-11-01411-f003:**
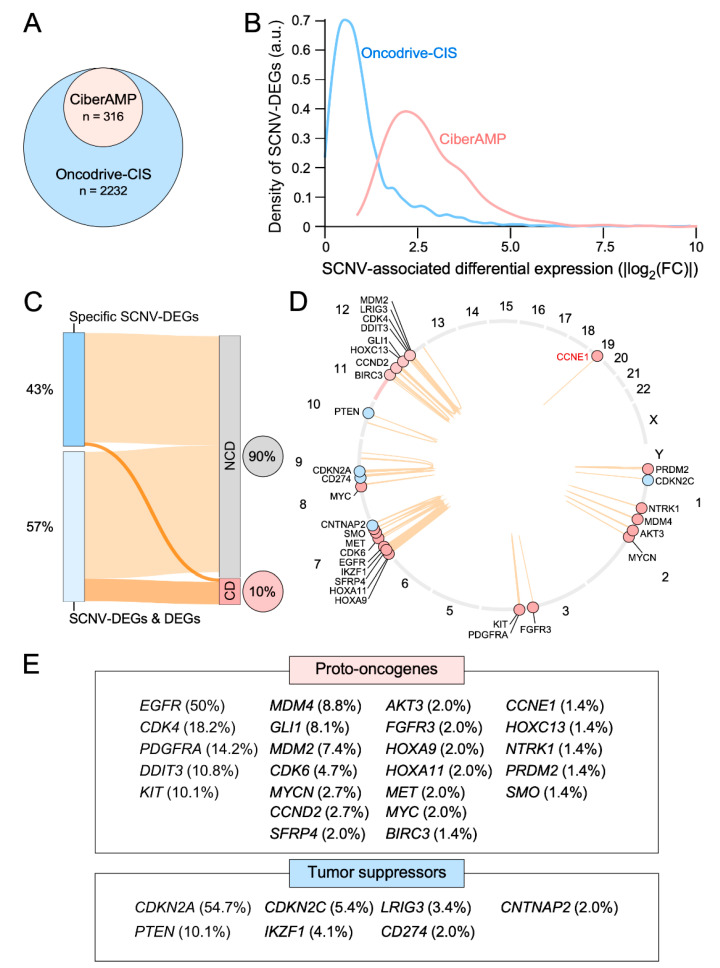
CiberAMP benchmarking. (**A**) SCNV-DEGs identified using the default searching parameters of CiberAMP and Oncodrive-CIS. a.u., arbitrary units. (**B**) Distribution of the SCNV-DEGs identified by CiberAMP (red) and Oncodrive-CIS (blue) according to the differential expression levels of the encoded transcripts. (**C**) Sankey diagram that summarizes the proportion of cancer drivers found among the 316 SCNV-DEGs identified by CiberAMP in glioblastoma. DEG, differential expression gene; CD, cancer driver according to the COSMIC Cancer Gene Census; NCD, non-cancer driver according to the COSMIC Cancer Gene Census. (**D**) Chromosomal localization (circle) and co-segregation pattern with other genomic regions (brown arches) of SCNV-DEGs that are associated with well-known proto-oncogenes (red circles) and tumor suppressor genes (blue circles). Chromosomes are indicated by the corresponding number outside the circle. (**E**) Proto-oncogene (top) and tumor suppressor (bottom) genes that undergo SCNV-DEG in glioblastoma according to CiberAMP analyses. The frequency of each SCNV-DEG in glioblastoma samples is included in brackets.

**Figure 4 biology-11-01411-f004:**
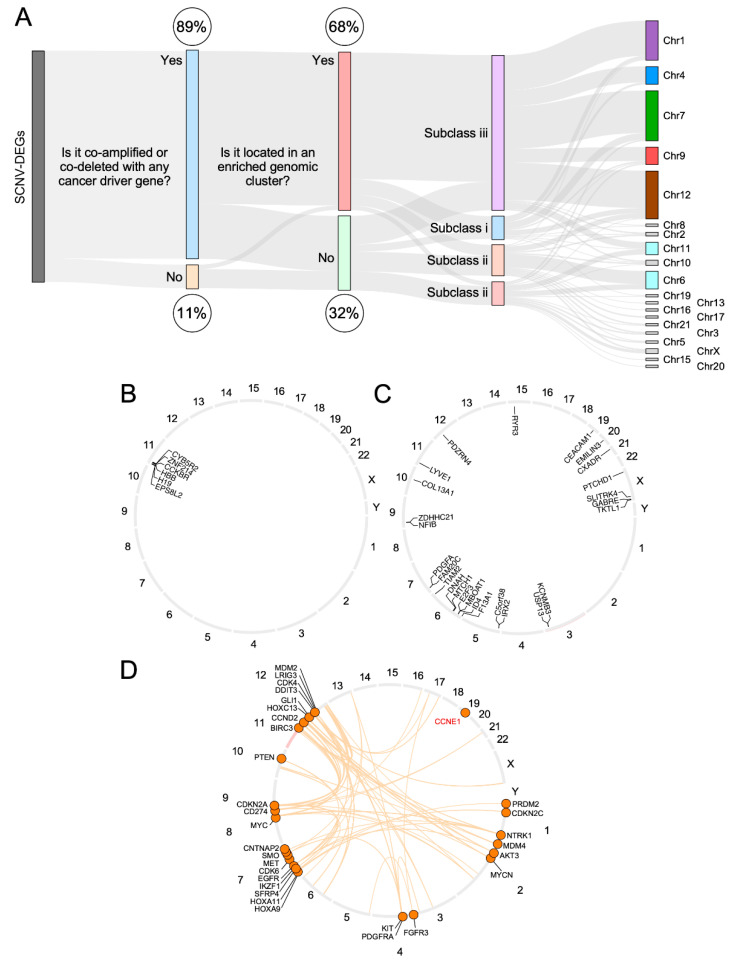
Functional cataloging of SCNV-DEGs in glioblastoma. (**A**) Sankey diagram showing the genomic context associated with CiberAMP-identified SCNV-DEGs in glioblastoma. Chr, chromosome. Subclass i, includes predicted drivers (SCNV-DEGs containing known proto-oncogenes or tumor suppressor genes); Subclass ii, includes two types of putative drivers: (a) SCNV-DEGs not associated with concurrent copy number variations in known cancer driver genes. (b) SCNV-DEGs concurrently amplified or deleted with SCNV-DEGs encoding cancer drivers located in different chromosomes; Subclass iii, includes SCNV-DEGs with uncertain functional relevance. See further information in the main text. (**B**–**D**) Genomic localization of indicated SCNV-DEGs that were cataloged as functionally interesting (subtype ii) according to our genomic context-based classification criteria (see main text for more details). Co-amplifications and co-deletions with these oncogenes are represented by arched brown lines.

**Figure 5 biology-11-01411-f005:**
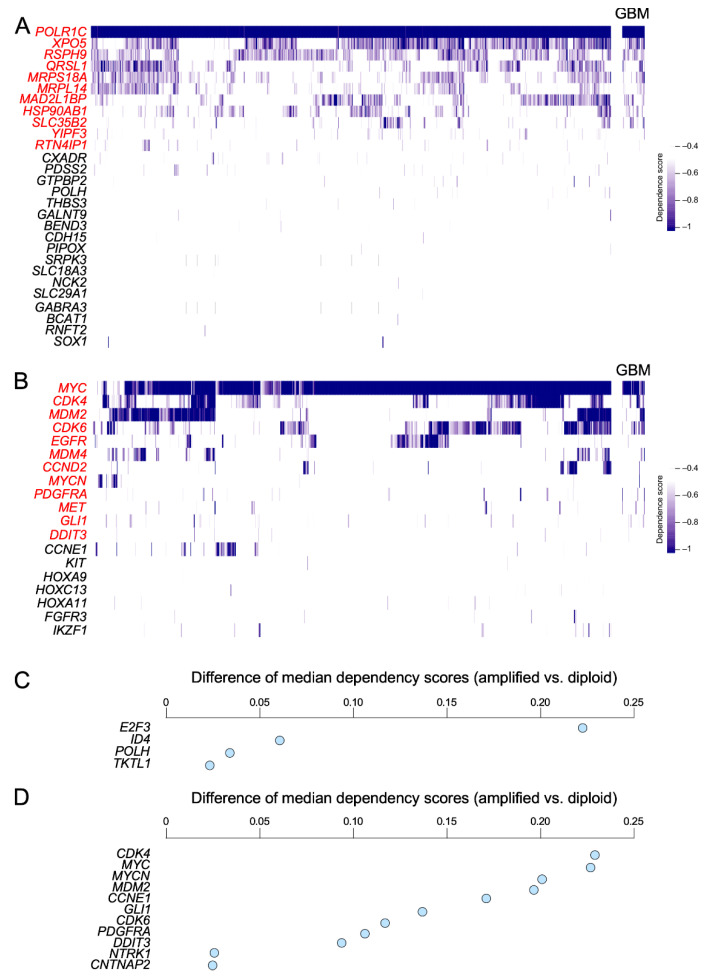
Validation of CiberAMP-identified SCNV-DEGs. (**A**) Effect of knockout of indicated genes related to SCNV-DEGs found in glioblastoma on the proliferation of cancer cell lines. In the right column (GBM), we specifically include genes whose knockout affects the proliferation of at least one glioblastoma cell line. The effect of proliferation is depicted as a blue gradient (see the scale on the right). Genes are shown on the left. In red, those that affect proliferation in at least one of the glioblastoma cell lines contained in the screening. Cell lines are shown in vertical lines (we have omitted the name of these lines for the sake of space). (**B**) Effect of the knockout of the indicated glioblastoma SCNV-DEGs harboring known cancer driver genes on the proliferation of cancer cell lines. Depiction of cell lines and proliferative effect of the gene knockout is shown as in (**A**). (**C**) Effect of knockout of the indicated genes linked to SCNV-DEGs found in glioblastoma on the proliferation of cancer cell lines that are positive for that SCNV. Depiction of cell lines and proliferative effect of the gene knockout is shown as in (**A**). (**D**) Effect of knockout of the indicated glioblastoma SCNV-DEGs harboring known cancer driver genes on the proliferation of cancer cell lines that are positive for that specific SCNV. Depiction of cell lines and proliferative effect of the gene knockout is shown as in (**A**).

**Table 1 biology-11-01411-t001:** Comparison between the main features of CiberAMP and Oncodrive-CIS.

Parameter	CiberAMP	Oncodrive-CIS
Language	R	Python
Input data	List of gene symbols and TCGA cohort IDs	Normalized RNA-seq and copy number data
Samples from healthy tissue	Yes	Yes
Score provided	Log_2_(FC) and adjusted *p* value	Combined score and adjusted *p* value
Use of validated pipelines for differential expression analyses	Yes	No
Provides log_2_(FC) values associated with the differential expression analyses	Yes	No
Provides information on differential gene expression between copy number altered and diploid tumor samples	Yes	No
Can analyze different tumor types in the same run	Yes	No
Provides information on concurrency or mutual exclusivity among SCNV-DEGs	Yes	No
Provides information on enriched SCNV-DEG genomic regions	Yes	No
Includes an interactive visualization tool to explore the outputs obtained from the analyses	Yes	No

**Table 2 biology-11-01411-t002:** CiberAMP benchmarking using indicated TCGA datasets. Please, note that the number of cancer drivers is progressively accumulated as we move from the 0–10% to the 90–100% bin. This latter bin contains all identified cancer drivers by each algorithm. In addition, note that the two algorithms use different criteria (e.g., overall impact on gene expression) to generate the list of SCNV-DEGs.

Bin	CiberAMP		Oncodrive-CIS	
	# Cancer Drivers	% Cancer Drivers	# Cancer Drivers	% Cancer Drivers
Glioblastoma				
0–10%	4	12.9	16	7.2
10–20%	10	15.9	33	7.4
20–30%	12	12.8	43	6.4
30–40%	15	11.9	50	5.6
40–50%	19	12.0	61	5.5
50–60%	22	11.6	71	5.3
60–70%	26	11.8	80	5.1
70–80%	27	10.7	92	5.1
80–90%	29	10.2	108	0.5
90–100%	31 (total)	9.8	119 (total)	5.3
Head and neck cancer				
0–10%	16	8.0	68	5.1
10–20%	29	7.3	120	4.5
20–30%	38	6.4	163	4.1
30–40%	44	5.5	219	4.1
40–50%	49	4.9	259	3.9
50–60%	53	4.4	315	3.9
60–70%	61	4.4	369	3,9
70–80%	65	4.1	424	3.9
80–90%	69	3.8	467	3.9
90–100%	76 (total)	3.8	514 (total)	3.9

## Data Availability

All data generated in this work as well as the source code of the algorithm are available at the GitHub repository: https://github.com/vqf/ciberAMP/ (accessed on 25 September 2022).

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
