# Peer review of "CiberAMP: An R Package to Identify Differential mRNA Expression Linked to Somatic Copy Number Variations in Cancer Datasets"

_biology, 2022, doi:10.3390/biology11101411_

Round 1

Reviewer 1 Report

The authors developed an R package named CiberAMP to detect SCNV-linked differentially expressed genes (SCNV-DEGs) using the public TCGA data. The algorithm and overall analysis presented in the manuscript seems reasonable and promising. I have one minor comment about the installation of the CiberAMP package. After trying the installation of the method, I noticed that this tool has a huge dependency on many other R packages in specific versions. The authors should clarify the complete list of R packages that are required within CiberAMP, and provide some user-friendly tutorials to install the tool smoothly and successfully. Please add this information in both Github page and the Supplementary Table.

Reviewer 2 Report

The current manuscript CiberAMP: An R Package to Identify Differential mRNA Expression Linked to Somatic Copy Number Variations in Cancer Datasets has some technical flaws and it requires significant major revision before it is accepted. 

Major revision: 

Author claims better accuracy of ciberAMP than the similar tool Oncodrive-CIS. I find several technical flaws in the result describing benchmarking.

1)    Author showed that “314 out of the 316 CiberAMP hits were also detected in the results obtained from Oncodrive-CIS” (line 305). On top of that Oncodrive-CIS also reports additional 1918 hits which are not reported by CiberAMP. Author stats large number of hits missed by CiberAMP is due to “presence of double fold-change and p-value thresholds to consider a SCNV-DEG as statistically significant in CiberAMP”. Additionally, author mentioned that “most of the extra SCNV-DEGs reported by Oncodrive-CIS are associated with marginal changes in transcript levels and associated with p-values much larger than 0.05 (Fig 3B)”. Figure 3B shows the comparison of fold change only and not the p-value. Therefore, it is not correct to mention “p-values” in the previous statement. In fact, it is important that author shows the p-value comparison like fold change. 

2)    The meaning of the phrase “double fold-change” is not clear in the statement: “presence of double fold-change and p-value thresholds to consider a SCNV-DEG as statistically significant in CiberAMP” 

3)    Interpretation of Figure 3B translates that, additional candidates reported by Oncodrive-CIS has “marginal changes in transcript levels” compared to CiberAMP. It is possible that even though targets have marginal changes, they could be biologically important. In fact, careful observation of Figure 3B suggest that average fold change for the majority of the transcripts reported by Oncodrive-CIS is >1.5 fold change (log2FC 0.58) which is widely accepted threshold for gene expression analysis. This suggest that most of the targets reported by Oncodrive-CIS are not with just marginal changes, instead they could be biological important targets and missed by CiberAMP. Hence, this observation questions the accuracy of CyberAMP.  

4)    I strongly recommend authors show benchmarking results in terms of sensitivity and specificity. 

5)    For fair comparison between two tools author must show a figure similar to 3C for the candidates derived by Oncodrive-CIS. 

6)    Author stats that “CiberAMP provides a list of hits more enriched in proto-oncogenes and tumor suppressor genes across all frequency intervals than Oncodrive-CIS”. It is not mentioned that how this enriched list of proto-oncogene and tumor suppressor genes have been derived. 

7)    Table 2 lists the % Cancer driver genes reported by CiberAMP and Oncodriver-CIS. Total number of cancer driver genes reported by CiberAMP and Oncodriver-CIS in glioblastoma are 195 and 673 respectively. Considering author’s statement -  “Another critical point in this type of algorithms is the ability to identify cancer relevant genes in the generated list of SCNV-DEGs in an unbiased manner” – number of oncogenes reported by such algorithms is one of the criteria to assess the quality of the algorithm.  This contradicts the results shown in manuscript as number of cancer relevant genes reported by Oncodriver-CIS is >3.5X higher than CiberAMP (Table 2). Hence, the sensitivity to detect cancer related genes by Oncodriver-CIS is more than CiberAMP. Same also applies to Head and neck cancer data shown in Table 2. 

8)    Analysis shown in figure 4A can be done for the candidate genes identified by Oncodriver-CIS to show how many hits can be grouped in to “Subclass i”, “Subclass ii and subclass iii”. 

9)    Author must submit R package to  R package repositories such as CRAN or Bioconductor to ensure the quality of code, documentation and installation on different computational platform such as linux, windows and macOS. It failed to install on my computer (macOS) suggesting it doesn’t meet to the required installation standards. See the screenshot below

Minor comments: 

10)  Manuscript has several grammatical and spelling errors. Some of them are 

a.     Line 45: “it is worth nothing that” … should be worth noting 

b.     Line 50: “wet lab” should be wet-lab. 

c.     Line 54: “during these last years” … doesn’t read well. 

d.     Line 66: “that a given gene can undergo between the reference normal tissue, the diploid 66 tumors, and the SCNV-positive ones.” …. Doesn’t read well. 

e.     Line 69 – 73: “To solve these problems, a second-generation bioinformatics tool (Oncodrive-CIS) was developed to integrate more adequately SCNV information with mRNA expression data using, in this case, a comparison of gene expression levels between SCNV-harboring tumor samples, diploid tumors and, in some cases, healthy tissue controls [20]”…..Sentence is too long to read in one go. 

f.      Line 141: “shallow or deep copy number variations, as well as the possibility to analyze them in a single group” … meaning shallow, deep and single group is not clear. 

g.     Line 450: “cell lines contained in that dabase into”.. database?

h.     Denotation of the word “p-value” is inconsistent and not scientific throughout manuscript. 

11) Reference missing - line 54  “international cancer genome projects” …

12) Line 138 – 139: “homozygous gene deletion (–2), gene hemizygous deletion (–1), diploid content (0), an extra copy of an allele (+1) and high-level gene copy gains (+2)”…… Meaning of values in parenthesis is not clear.

13) Line 182: “The other plots found in this article are created using other R packages and no function is provided with CiberAMP to make them”… Author didn’t acknowledge other R packages as none of them have been cited. All R packages used must be cited.   

14) Table 1:

a.     Row number 6 and 7 seems redundant. If not, elaborate each. 

b.     Table 1: Row number 9 “It supports analysis of multiple tumor types at the same time”… meaning of same time is not clear. 

15) Line 267: “high level SCNVs” … How to interpret “high level” is not clear.  

16) Figure 1: Figure legends are not accurate. 

a.     The legend “SCNV-DEG type” is misleading. Since all dots in the figure denotes SCNV-DEGs labelling one of the categories as “SCNV-DEG” (the bottom one) is technically incorrect.

b.     The legend SCNV frequency (%) must be shown without any colors. 

c.     Figure legend doesn’t explain meaning of subcategories (Subset a.1, Subset b.1, Subset c.1 etc…) within each subset. 

d.     Subset b.1 is highlighted by two UP arrows each with different shades of red color. Legend describing red shade of bottom arrow is missing. Same for subset b.2. 

e.     In the figure gene BIRC3 seems wrong labelled as the color of the dot is different than other proto-oncogenes.

17)  Line 306: meaning “X%” is not clear.  

18) Line 320: “SCNV-DEGs in an unbiased manner”… Meaning of unbiased is not clear. 

19) Author must describe the meaning of “Subclass i”, “Subclass ii and subclass iii” in the legend of Figure 4. 

20) Line 330: The phrase “driver-associated” used to reveal genes which are cancer driving genes. However, the phrase “driver-associated” doesn’t describe meaning well. Furthermore, same genes described by the phrase “cancer drivers” in Table 2. I suggest author to be consistent for the choice of the words used to describe the same thing throughout manuscript.

Round 2

Reviewer 2 Report

The manuscript has improved significantly after the first revision. It can be accepted for publication.